# Deterministic Tractography Analysis of Rat Brain Using SIGMA Atlas in 9.4T MRI

**DOI:** 10.3390/brainsci11121656

**Published:** 2021-12-18

**Authors:** Sang-Jin Im, Ji-Yeon Suh, Jae-Hyuk Shim, Hyeon-Man Baek

**Affiliations:** 1Department of Core Facility for Cell to In-Vivo Imaging, Lee Gil Ya Cancer and Diabetes Institute, Gachon University, Incheon 21999, Korea; toupgh94@gmail.com (S.-J.I.); grassyear@gmail.com (J.-Y.S.); 2Department of BioMedical Science, Lee Gil Ya Cancer and Diabetes Institute, Gachon University, Incheon 21999, Korea; jaehyukshim11@gmail.com; 3Department of Molecular Medicine, Lee Gil Ya Cancer and Diabetes Institute, Gachon University, Incheon 21999, Korea

**Keywords:** tractography, diffusion tensor image, rat brain, segmentation, atlas

## Abstract

Preclinical studies using rodents have been the choice for many neuroscience researchers due totheir close reflection of human biology. In particular, research involving rodents has utilized MRI to accurately identify brain regions and characteristics by acquiring high resolution cavity images with different contrasts non-invasively, and this has resulted in high reproducibility and throughput. In addition, tractographic analysis using diffusion tensor imaging to obtain information on the neural structure of white matter has emerged as a major methodology in the field of neuroscience due to its contribution in discovering significant correlations between altered neural connections and various neurological and psychiatric diseases. However, unlike image analysis studies with human subjects where a myriad of human image analysis programs and procedures have been thoroughly developed and validated, methods for analyzing rat image data using MRI in preclinical research settings have seen significantly less developed. Therefore, in this study, we present a deterministic tractographic analysis pipeline using the SIGMA atlas for a detailed structural segmentation and structural connectivity analysis of the rat brain’s structural connectivity. In addition, the structural connectivity analysis pipeline presented in this study was preliminarily tested on normal and stroke rat models for initial observation.

## 1. Introduction

Rats are the experimental subjects of choice for many preclinical studies due to their close reflection of human biology [1,2]. In particular, preclinical studies using rodents play an important role in coordinating brain anatomy and function in neuroscience research [3].

Studies have used magnetic resonance imaging (MRI), which exhibits high spatial resolution and contrasts to accurately identify and delineate brain regions [4,5,6]. In addition, tractographic analysis using diffusion tensor imaging (DTI), a method that utilizes diffusion to construct fibers representing the neural structure of white matter, have been used to identify partially altered neural connections that can contribute to various neurological and psychiatric diseases [7,8,9]. In order to identify cortical regions related to specific functions, studies have used diffusion tractography to determine the connectivity network between regions with common functional features. Through a selective tracking method, we investigated the relationship between brain structure and function.

The method of generating tractography is typically divided into a deterministic method and a probabilistic method, depending on the fiber orientation sampling method used for tractography propagation. For probabilistic tractography, fiber orientation distribution is estimated for each voxel and randomly extracted from the distribution to determine the streamline propagation direction [10], whereas the deterministic tractographic method extracts streamlines in a fixed direction at each voxel [11]. It is known that deterministic tractography methods cannot account for the inherent uncertainty of fiber orientation estimates and are sensitive to ambient principal orientation and noise; probabilistic tractography, on the other hand, can quantify the probabilistic reliability of each reconstructed path, accounting for uncertainty in data [12,13,14]. Although probabilistic methodology has been considered to be the better method for reconstruction and tractography, it is likely that the certainty of reconstructed fibers are less and less significant for applications such as connectome mapping due to the multiplicity of tests [15,16,17]. Additionally, a recent study comparing tractographic algorithms reported that deterministic tractographic methods can sometimes outperform probabilistic methods [18,19,20,21].

For image data analysis, there are templates that are universally used in humans, and various atlas and brain segmentation schemes have been created to fit such templates [22,23,24,25,26]. However, there is no scientific consensus on how to analyze image data and atlas-based neuroinformatics in Rat studies [3]. Because every brain has a unique volume and shape, standardized anatomical templates and spaces that enable spatial normalization of data and co-mapping of empirical effects are needed for comprehensive analysis. In addition, a standardized atlas capable of identifying anatomically segmented regions of interest (ROIs) can be beneficial for normalized comparisons of different studies [27,28,29].

In this study, we present a comprehensive methodology using detailed structural segmentation of the rat brain made possible by using the SIGMA atlas for deterministic tractographic analysis of structural connectivity based on the segmented structural region. We applied our methodology presented in our study to the brains of normal Rat and stroke Rat models.

## 2. Materials and Methods

### 2.1. Preparation of Animals

Analysis methods were applied on one normal rat and six middle cerebral artery occlusion (MCAO) model rats. Three-month-old male Sprague-Dawley rats (SD, Orient Bio, Seoul, Korea) weighing 250–350 g were used in this experiment. All rats were bred in transparent cages, one to two according to their body weight, and exposed to light and darkness for 12 h each day. The temperature of the rat cage was maintained at 21 to 24 °C. All animal experiments and procedures were performed according to the guidelines of the Association for Assessment and Accreditation of Laboratory Animal Care International (AAALAC International, www.aaalac.org accessed on 10 December 2021), and the Center of Animal Care (CACU, Center of Animal Care and Use, Lee Gil Ya cancer and Diabetes Institute, Gachon University, Incheon, Korea) approved and processed the animal test protocol (LCDI-2020-0105).

### 2.2. Animal Models

The animal model of middle central cerebral artery occlusion (MCAO) induced by focal cerebral infarction was performed using a previously known method [30]. SD rats were anesthetized with 1.5 to 2% isoflurane, the right common carotid artery was exposed, and the external and internal carotid arteries were separated.

After incision of the external carotid artery, which is 7 to 8 mm from the bifurcation, the external carotid artery was placed in a straight line with the internal carotid artery. A 4–0 black monofilament suture coated with silicone (diameter: 35 µm) was inserted to the puncture site of the external carotid artery (403656PK10, Doccol Cooperation, Sharon, MA, USA) and moved toward the origin of the middle cerebral artery by passing through the internal carotid artery. After 90 min of MCAO, the inserted filament was retrieved for the reperfusion of cerebral blood flow. During the surgery and recovery period, the temperature was adjusted to 37.0 ± 0.5 °C with a thermostat using a heating pad (Thermo Fisher Scientific, Waltham, MA, USA).

### 2.3. MRI Acquisition

Image data acquired in this study were performed on a 9.4T Bruker BioSpec horizontal bore animal scanner (Bruker Biospin, Ettlingen, Germany) equipped with a tilt system of (660 mT/m). The image data collection of normal rats was performed at the Core facility for Cell to In-vivo Imaging (CII, Gachon University, Lee Gil-ya Cancer Diabetes Research Institute), and the image data collection of the MCAO model was performed at Sungkyunkwan University N Center (IBS, institute for Basic Science, Suwon, Korea). A quadrature volume resonator (inner diameter (114 mm); Bruker Biospin, Ettlingen, Germany) was used for RF excitation and a four-channel mouse brain surface coil (Bruker Biospin, Ettlingen, Germany) was used for signal reception, and the software was Paravision 6.0. The pulse sequence used for this acquisition was a 2D EPI-diffusion tensor, with a normal Rat (Spin echo sequence with a repetition time = 2500 ms, echo time = 21.3165 ms, flip angle = 90°, bandwidth = 170 kHz, b-value = 2011.85 s/mm², diffusion gradient pulse duration (δ) = 4.5 ms, diffusion gradient separation (Δ) = 10.6 ms, diffusion direction = 30, field of view = 2.5 × 3.5 cm^2^, slice thickness = 0.4 mm, matrix = 125 × 175, slice = 40, resolution = 200 × 200 × 400 µm^3^, four averages and resulting in a total acquisition time of 1 h 15 m 50 s) and modeling Rat (Spin echo sequence with a repetition time = 3000 ms, echo time = 17.0505 ms, flip angle = 90°, bandwidth = 341 kHz, b-value = 1389.93 s/mm², diffusion gradient pulse duration (δ) = 2.5 ms, diffusion gradient separation (Δ) = 8.5 ms, diffusion direction = 30, field of view = 2.5 × 2.5 cm^2^, slice thickness = 0.3 mm, matrix = 83 × 83, slice = 115, resolution = 301 × 301 × 300 µm^3^, two averages and resulting in a total acquisition time of 28 m) were scanned. The signal-to-noise ratio (SNR) of the acquired DTI data was 25 in Normal rats, and the DTI SNR of stroke models were one day old = 16, one week old = 18, two weeks old = 17, four weeks old = 18, six weeks old = 20, and seven weeks old = 19. DTI data FA maps of normal rat and stroke rat models with different parameters are presented in Appendix A.

### 2.4. Image Data Processing

The acquired DTI data was first processed via ANTx2 (Atlas Normalization Toolbox using elastix 2, University Medicine Berlin, Berlin, Germany) [31,32,33]. Data in Bruker format was converted to Neuroimaging Informatics Technology Initiative (NIFTI) format and normalized to SIGMA space. B0 images were extracted from the normalized image data with the skull. Brain and brain structure masks were acquired by segmenting each ROI used for analysis on the extracted b0 image data. The acquired ROI masks were registered to the DTI data using the FMRIB software library version 6.0.2 (FSL, created by the Analysis Group, Oxford, UK). After all the masks were linearly registered to DTI space using the FLIRT function, the registered masks were qualitatively evaluated according to whether each mask was registered to the correct position [34,35]. MRtrix3 was used for the preprocessing of the DTI data. DTI data was denoised and bias field corrected to remove noise and correct for B1 field non-uniformity [36]. Additionally, FSL’s eddy correct was used to correct for distortions and motion artifacts [37]. The preprocessed data were used for deterministic tractography analysis in DSIsudio (http://dsi-studio.labsolver.org/ accessed on 15 December 2021). After the DTI data was converted into SRC format, a range of brain voxels specified by the segmented masks was selected for fiber orientation reconstruction and fiber tracking. The analysis pipeline is presented in Figure 1.

### 2.5. Deterministic Tractography

Tractography of ROIs were obtained using DSIstudio’s Q-sampling imaging (GQI), which involves decomposing up to two fibers in one voxel by a wireline tracking algorithm. A deterministic streamlined tracking algorithm with high connectivity validation was reconstructed through each ROI selected from the SIGMA atlas (Primary motor cortex, M1; Secondary Motor Cortex, M2; Primary Somatosensory Cortex, S1; Secondary Somatosensory Cortex, S2; Corpus Callosum and Associated Subcortical White Matter, Corpus Callosum and Associated Subcortical White Matter, CC; Internal Capsule, IC; Cerebral Peduncle, CP). Each reconstructed fiber that passes through the ROI, enters that region and does not proceed further. The fiber tracking (Tracking Threshold: 0.1, Angular Threshold: 45°, Step Size: 1.5, Min Length: 0.5, Max Length: 250, Terminate if: 2,000,000) process results in the number and shape of the streamlines passing through the target area

## 3. Results

### 3.1. SIGMA Atlas-Based Whole Brain Segmentation and Registration

Segmentation of whole brain structural regions of rats using the SIGMA atlas was performed on B0 data by registering both data and atlas to accurately overlap. A cross-section of the SIGMA atlas used in the study is presented in Appendix A. In order to qualitatively verify the accurate segmentation information of detailed structures, the segmented brain structure region on the B0 image data is visualized in Figure 2. The division and registration of all structures appears to be clearly registered at each location based on the SIGMA atlas, and it can be confirmed that even strong deformations of anatomical structures are outlined realistically by the algorithm. The segmentation and registration results are 3D rendered and presented in Figure 3 so that location information and shapes can be checked from various directions. The names and abbreviations and color codes of all visualized brain structural regions are presented in Appendix A.

### 3.2. Deterministic Tractographic Analysis

For structural connectivity, seven regions (M1, M2, S1, S2, CC, IC, CP) related to the corticospinal tract (CST) that transmit movement-related information from the cerebral cortex to the spinal cord were segmented and registered. The Volume, Intensity, FA and MD values of each structure obtained in the process of processing the DTI data are presented in Appendix A. All segmented and registered structural regions were subdivided into the left and right hemispheres, and connectivity between the 14 regions was generated. Each structural region was divided into seed and target, and deterministic tractographic analysis was performed, with the results indicating the connectivity between each structural region as presented in Figure 4 and Figure 5.

Figure 4A shows a 3D rendering of a rat’s brain and each structural region, providing localization and visualization of connectivity strength between each region. Connectivity strength between the 14 structural areas is represented by white lines. The stronger the connection strength, the thicker the white line, and the weaker the connection, the thinner the white line. In Figure 4A, the strength of the connection from each structure location to another structure area in the rat’s brain can be visualized. Furthermore, the connection strength between structural regions is presented in matrix form in Figure 4B for direct comparison. The connectivity between all structural regions was represented as a color map, with colors closer to red indicating stronger connectivity and colors closer to blue representing weaker connectivity. The left row of the matrix represents the seed area and the top row represents the target. In addition, the location information of each structural region are presented in 3D renderings. Figure 5 presents a connectogram plot showing the strength of the connections in each structural region. M1, M2, S1, S2, CC, IC and CP used in the study were divided into left and right hemispheres, with the connectivity between the 14 regions cross-validated and identified. 

The connectivity analysis between each structural area showed the connectivity of each hemisphere in the left and right hemispheres. The connectivity between structural regions that were close together showed strong connectivity, while regions farther away showed weaker connectivity. Interconnectivity of structures in the same hemisphere showed higher connectivity strength than connectivity in different hemispheres, and connectivity in different hemispheres showed weak or no connectivity. The connection strength was found to be mostly higher in the left hemisphere than in the right hemisphere. The association values between each structural region through tractographic analysis are presented in Appendix A.

### 3.3. Application of Deterministic Tractographic Analysis of Stroke Model

The segmentation and deterministic tractographic analysis pipeline established in this study was applied to the stroke model. In the stroke Rat model, images were acquired once every day, at week, at two weeks, at four weeks, at six weeks, and seven weeks after the onset of the disease, and disease progression was observed. Figure 6 investigates structural connectivity in the motocortex of the left and right hemispheres known to suffer from stroke. Figure 6A shows the neuronal pathways between Left M1 and Right M1 in unmodeled normal rats, and Figure 6B shows the changes in the neuronal pathways between Left M1 and Right M1 from day one to seven weeks after stroke onset.

It was confirmed that the Left M1-Right M1 connection of the stroke model was significantly reduced on the first day compared to the normal, and it recovered gradually and showed a shape similar to that of the normal rat after the secondweek. The left M1 and Right M1 connection strength values of the normal rat and stroke rat models are presented in Appendix A.

## 4. Discussion

Preclinical studies using MRI provides an opportunity to investigate various influences on the structural and functional aspects of the brain, such as behavioral, anatomical, physiological, biochemical and pathological analyses [38,39,40]. In particular, the research method using MRI has been used in many studies because it is non-invasive, can have high throughput, and has strong reproducibility [41,42,43,44]. However, unlike image analysis studies of humans in which various programs have been developed and validated, the standardization of image data acquisition, processing, and sharing using MRI in preclinical rat research requires a diverse evaluation process [45,46].

We present a tractographic analysis pipeline that can determine the segmentation of detailed structural regions and connectivity based on structural regions using MRI image data of the rat brain. The pipeline efficiently combines a variety of existing neuroimaging analysis tools to enable structural segmentation and tractographic analysis. In addition, the entire brain template of the rat is provided using the highly accurate SIGMA atlas from which a researcher can acquire detailed structural region information through segmentation, as well as perform regional analysis of image data [29].

All acquired image data were strictly registered as data in SIGMA space via ANTx, ensuring parallelism to the coronal plane of the atlas [32,33]. In addition, the results obtained from the pipeline were overlaid on the B0 image and qualitatively evaluated. However, since it is not possible to directly check the registration between the data sets of the divided regions, it is difficult to compare the volumes of the registered regions with the real rat brain structure region. As a result, careful evaluation of the registration method is required to confirm direct accuracy. However, for the segmented structures in this analysis, we were able to confirm that each brain structure region was accurately registered by performing strict registration based on the SIGMA atlas [47].

By applying a pipeline to the acquired DTI data, we were able to successfully obtain the results of connectivity analysis between structural regions. We were able to quantitatively check the connectivity between each structural region, and generate 3D renderings of the connectivity strength, connectivity matrix, and connectivity pathways for comparison.

The tractographic analysis, used to quantify connectivity between each structural region, showed low connectivity strength of interhemispheric structures, while intrahemispheric structures showed higher connectivity strength, similar to the results of previous studies. In addition, connectivity between structural regions close to each other was stronger than connectivity between structural regions further away from each other [48,49]. This difference in connectivity between structural regions was confirmed not only in normal rats but also in stroke rat models, similar to the results of previous studies. A previous study showed that neural connectivity in the damaged brain region initially weakened but recovered over time in rat stroke models, which were constructed by inducing localized unilateral damage limited to a portion of the sensorimotor cortex without directly damaging the CST [50,51]. Another study identified local changes in diffusion parameters at the site of sciatic nerve injury, and at locations both proximal and distal to nerve injury [52].

DTI, which can characterize the orientation and integrity of white matter, has been widely used in preclinical neurological studies for its ability to reconstruct white matter track pathways, and derive diffusion parameters which are particularly useful for the diagnosis and characterization of brain diseases [53,54]. DTI analysis has been popular for neuroscientists to identify brain connectivity and quantify tractographic-derived connectivity strengths between brain structural regions [55,56,57]. In previous studies, excellent performance results of tractography were reported when comparing neuroanatomical tracer data and tractography [58]. However, since data obtained with fiber tracing have potential pitfalls and limitations, great care must be taken when interpreting results [59,60,61,62]. To minimize such errors in DTI analysis, various studies have continuously made attempts to identify limitations and causes of errors in tractography, then incorporate potential methods for improving results [63,64]. Despite these challenges, data from tractography can provide important insightsinto anatomical connectivity.

In this study, we presented a structural analysis and structural connectivity analysis pipeline of the rat brain by combining various existing neuroimaging analysis tools. From the results, we were able to successfully extract and segment individual ROI masks and perform tractographic analysis. However, the study has several limitations. Due to the small number of rat samples used in the experiment, it was not possible to statistically verify the quantitative values obtained for each analysis. In the future, there is a need to increase the number of samples coupled with a longitudinal study. In addition, directly comparing pipeline results obtained using normal and stroke models may be difficult because the image data acquisition parameters of the normal rat and stroke rat models are different. In future studies, all images of both normal and stroke models will be obtained using the same parameters. Another limitation lies in the registration method used to obtain the desired structural region mask in the pipeline presented in the study. It is difficult to confidently measure volumetric change of mouse brain structures, due to possible errors that may have occurred due to the quality of the acquired image or the registration algorithm. In addition, there is a limitation to the careful evaluation of the registration method used to measure accuracy. In this study, although the SIGMA atlas has been used previously for accurate structural segmentation of the rat brain, there are limitations to how accurate the registration tools and qualitative visual inspection can be. Finally, the results of tractographic analysis can be influenced by various factors such as the quality of the image data and the parameters used to trace the fiber. For example, due to the low image resolution, only large clumps of axons with uniform orientation were studied, partially due to the fact that tensor-based calculations used for this study showed no mixture of curvature or fiber orientation within the pixel. Low SNR of our images may have caused errors in diffusion reconstruction, due to the high level of noise and partial volume effects [65,66]. It is also difficult to interpret certain characteristics of fibers because certain aspects of the generated fibers such as the projection direction cannot be distinguished.

## 5. Conclusions

In this study, we performed structural connectivity analysis through structural region segmentation using rat diffusion tensor images. We present a deterministic tractography analysis pipeline based on the structurally accurate SIGMA atlas for reconstructing connectivity between rat brain structures, which was preliminarily used to identify initial differences in the connectivity of normal and stroke model rats. The pipeline presented in this study can contribute to standardizing various data types and analysis methods in the field of neuroscience using preclinical animals, which can enable comprehensive application of structural analysis and structural connectivity.

## Figures and Tables

**Figure 1 brainsci-11-01656-f001:**
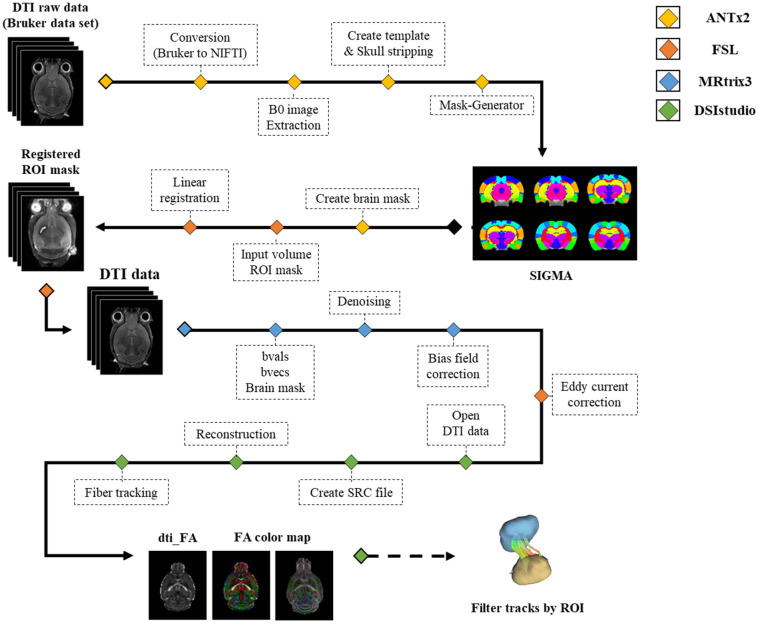
All analysis pipeline of the image data. Abbreviation: ANTx2, Atlas Normalization Toolbox using elastix 2; FSL, FMRIB software library; ROI, Region Of interest; FA, Fractional anisotropy; DTI, Diffusion tensor Image; NIFTI, Neuroimaging informatics technology initiative.

**Figure 2 brainsci-11-01656-f002:**
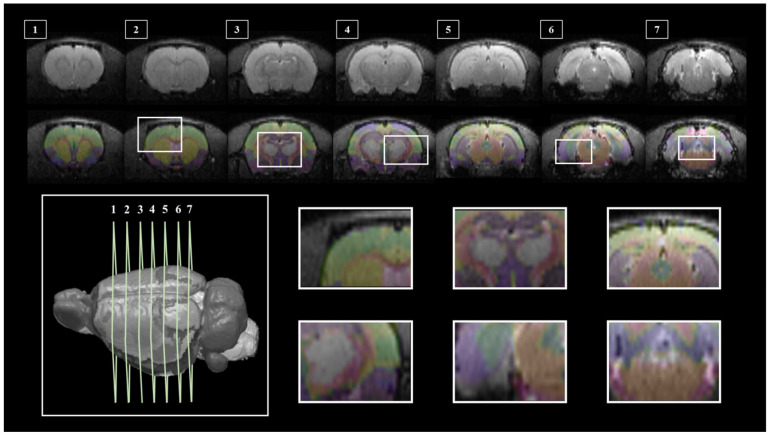
SIGMA Atlas-based Whole Brain Segmentation and Enrollment Results. Slices (**1**–**7**) shows the location of the segmentation result overlaid on the image data, 3D rendered RAT brain (**bottom left**) to show each slice location, and six detailed views show atlas overlays (**bottom right**).

**Figure 3 brainsci-11-01656-f003:**
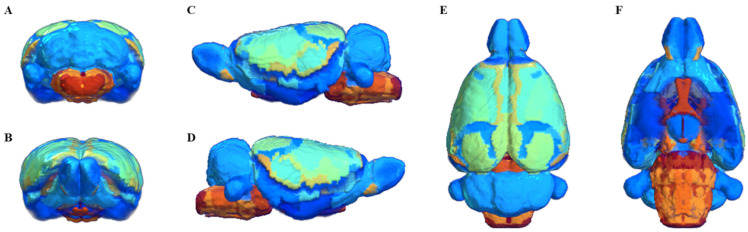
3D rendering of segmentation and registration results. posterior (**A**), anterior (**B**), left lateral (**C**), right lateral (**D**), superior (**E**), and inferior (**F**). Regions are colored to identify their boundaries, and color similarity between spatially separated regions is meaningless.

**Figure 4 brainsci-11-01656-f004:**
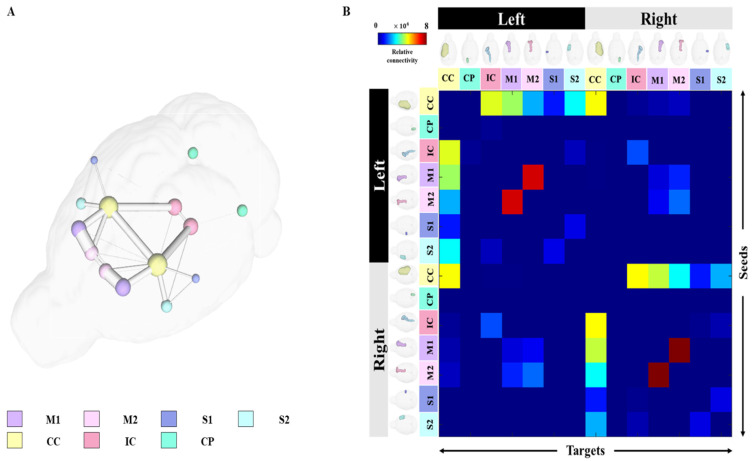
Results of tractography analysis between 14 anatomical regions. The connectivity between each structural region in the 3D rendered whole brain is shown in (**A**), and the label of each structural region is shown at the bottom of the figure. In addition, the connectivity matrix between each structural region was plotted using (**B**) connectivity strength color maps with the seed plotted on the left and the target plotted on top. Abbreviations: Primary motor cortex, M1; Secondary Motor Cortex, M2; Primary Somatosensory Cortex, S1; Secondary Somatosensory Cortex, S2; Corpus Callosum and Associated Subcortical White Matter, Corpus Callosum and Associated Subcortical White Matter, CC; Internal Capsule, IC; Cerebral Peduncle, CP.

**Figure 5 brainsci-11-01656-f005:**
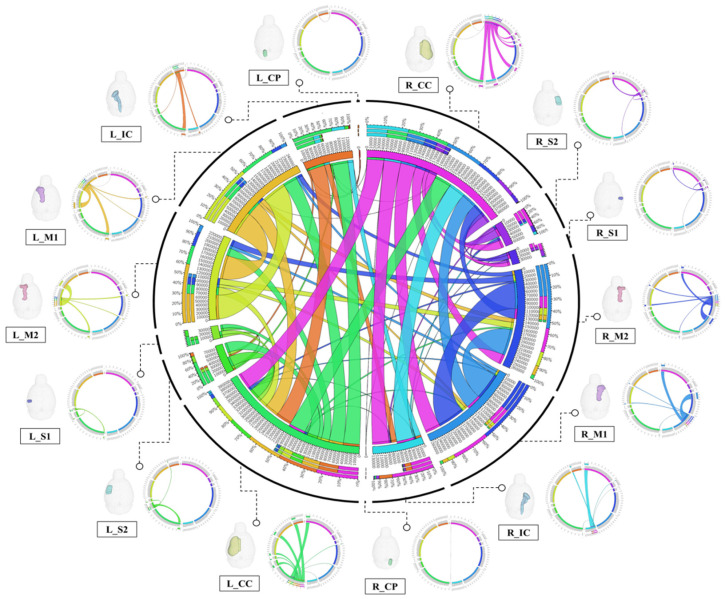
Connection diagram showing the structural connectivity of the consensus among the 14 structural domains. The structural regions of each cluster are represented by rectangles around a large circle, and the lines connecting the rectangles indicate the connections between the corresponding structural regions. The thicker the line, the higher the connectivity, and the thinner the line, the lower the connectivity. The center circle is expressed as the sum of the strengths of the connections between all structural regions, and the diagrams from each structural region (seed) to the other structural regions (target) are expressed around the circle. Each seed area was 3D rendered to represent the exact location, and lines from the specified area to all targets were individually rendered.Abbreviations: Primary motor cortex, M1; Secondary Motor Cortex, M2; Primary Somatosensory Cortex, S1; Secondary Somatosensory Cortex, S2; Corpus Callosum and Associated Subcortical White Matter, Corpus Callosum and Associated Subcortical White Matter, CC; Internal Capsule, IC; Cerebral Peduncle, CP; Left, L; Right, R.

**Figure 6 brainsci-11-01656-f006:**
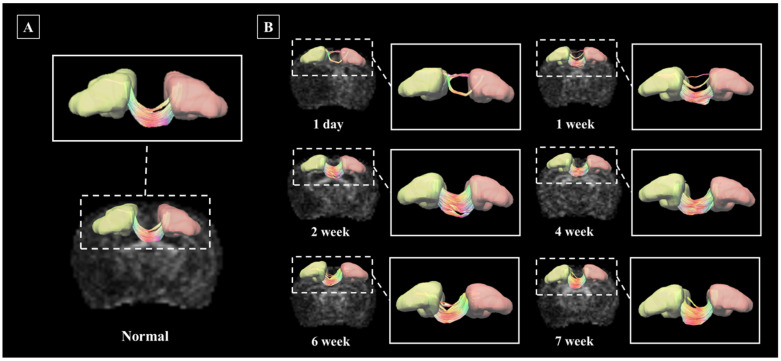
Nerve pathways in the Left M1 and Right M1 regions, shown in D rendering. Linkage pathways of normal rats (**A**), and linkage pathways by date of disease occurrence in a stroke rat model (**B**). Rendered structures and connection paths are represented as axial planes, and connection paths between structural regions are enlarged and presented in greater detail.Abbreviations: Primary motor cortex, M1.

## Data Availability

Data are available for replication upon request.

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
