# Peer review of "Deterministic Tractography Analysis of Rat Brain Using SIGMA Atlas in 9.4T MRI"

_brainsci, 2021, doi:10.3390/brainsci11121656_

Round 1

Reviewer 1 Report

The paper presents a method that allows obtaining structural segmentation of the rat brain and showing structural connectivity based on the segmented regions. The paper is well written, presented results are convincing. The main concern, however, is related to the contribution and the novelty of this work. Tractography is well established technique, enabling the reconstruction of the course of nerve fibers in a three-dimensional color projection in accordance with the greatest diffusivity, running in the brain from one region to another. Thus, results presented by Authors can be obtained for a long time, thanks to the appropriate software. Therefore, please convince the reader of the novelty of the applied approach in the light of the existing methods described in the literature. What problems related to the use of the tractography have been eliminated / removed thanks to the method used?

Other comments:

  1. What is the accuracy of the segmentation of the brain structures shown in Fig. 2? How was this accuracy verified?
  2. How the imaging results are influenced by the limitations of the tractography method, e.g. a voxel may contain tens of thousands of axons of different orientations or nerve fibers of similar size but oriented in opposite directions may be present next to each other (no path will be visualized in this case).
  3. The Conclusion section is missing in this paper.
  4. The number and date of the decision of the ethics committee approving the the animal test protocol were not provided.

Author Response

Response to Reviewer 1 Comments

Point 1: The paper presents a method that allows obtaining structural segmentation of the rat brain and showing structural connectivity based on the segmented regions. The paper is well written, presented results are convincing. The main concern, however, is related to the contribution and the novelty of this work. Tractography is well established technique, enabling the reconstruction of the course of nerve fibers in a three-dimensional color projection in accordance with the greatest diffusivity, running in the brain from one region to another. Thus, results presented by Authors can be obtained for a long time, thanks to the appropriate software. Therefore, please convince the reader of the novelty of the applied approach in the light of the existing methods described in the literature. What problems related to the use of the tractography have been eliminated / removed thanks to the method used? 

 Response 1: Thank you for your kind review.

We believe that precisely segmenting and identifying structural regions of the brain is the most fundamental step, with tractography for identifying structural connectivity.

Brain studies with MRI typically utilize well-established atlases to acquire regions of brain structures by registering one or more pre-labeled atlases using a deformable registration method to subject volumes. [1-3].

In contrast to human brain MRI analysis studies that can utilize an immense library of well-established atlases and validated softwares rat MRI studies have much less to rely on as there is less interest in developing atlases and segmentation tools for small animals [4,5].

The methodology of registering atlas information to a subject is highly dependent on the accuracy of the atlas [6]. Therefore, utilizing an accurate atlas such as the SIGMA rat atlas [10] for segmenting rat brain structures using software such as ANTx2 and FSL’s FLIRT [7-9,11,12] can ensure accurate segmentations of rat MRI images. In this study, we also show that SIGMA atlas segmentations done using ANTx2 and FLIRT on normal and stroke model rat MRI images can be coupled with DSI studio for diffusion tractography analysis.

Point 2: What is the accuracy of the segmentation of the brain structures shown in Fig. 2? How was this accuracy verified?

Response 2: Determination of image registration accuracy is known to be non-standardized or not well-defined, and registration results are usually qualitatively evaluated visually [13].

In this study, after acquiring segmenting the high-accuracy SIGMA atlas through ANTx2 software, five researchers repeatedly performed registration tasks using FSL FLIRT, a validated registration method, then visually inspected the brain structure segmentation results.

Split validation was evaluated qualitatively, and limitations were drawn up at the conclusion of the Discussion.

Point 3: How the imaging results are influenced by the limitations of the tractography method, e.g. a voxel may contain tens of thousands of axons of different orientations or nerve fibers of similar size but oriented in opposite directions may be present next to each other (no path will be visualized in this case). 

 Response 3: The following content has been added to the Discussion part by collecting the opinions of reviewers.

“Tractography results can reveal errors in preprocessing such as image acquisition, local voxel-by-voxel reconstruction, and voxel-to-voxel tracing. Errors occur particularly during image acquisition and using EPI for DTI acquisitions may cause noise and artifacts due to sensitivity gradients that affect motion and eddy currents. In addition, millimeter-scale voxels containing thousands of axons with many complex configurations can present problems in reconstruction algorithms, resulting in inaccurate results [59]. In particular, it can be seen that deterministic Tractography methods cannot account for the inherent uncertainty of fiber orientation estimates and are sensitive to ambient principal orientation and noise [12].”

The corrected part is marked in brown.

Point 4: The Conclusion section is missing in this paper.

Response 4: A Conclusion section has been added to the revised Draft based on feedback from reviewers. The corrected part is marked in brown.

Point 5: The number and date of the decision of the ethics committee approving the the animal test protocol were not provided. 

Response 5: The contents of the animal experiment protocol were recorded in the Institutional Review Board Statement before the references section. The corrected part is marked in brown.

[1] K. K. Leung, K. K. Shen, J. Barnes, G. R. Ridgway, M. J. Clarkson, J. Fripp, O. Salvado, F. Meriaudeau, N. C. Fox, P. Bourgeat, S. Ourselin, "Increasing power to predict mild cognitive impairment conversion to Alzheimer’s disease using hippocampal atrophy rate and statistical shape models," In International Conference on Medical Image Computing and Computer-Assisted Intervention. Springer, Berlin, Heidelberg, pp. 125-132, 2010.

[2] M. J. Cardoso, K. Leung, M. Modat, S. Keihaninejad, D. Cash, J. Barnes, N. C. Fox, S. Ourselin, Alzheimer’s Disease Neuroimaging Initiative, "STEPS: Similarity and Truth Estimation for Propagated Segmentations and its application to hippocampal segmentation and brain parcelation," Medical image analysis, Vol. 17, No. 6, pp. 671-684, 2013.

[3] D. Ma, M. J. Cardoso, M. Modat, N. Powell, J. Wells, H. Holmes, F. Wiseman, V. Tybulewicz, E. Fisher, M. F. Lythgoe, S. Ourselin, "Automatic structural parcellation of mouse brain MRI using multi-atlas label fusion," PloS one, Vol. 9, No. 1, 2014.

[4] D. A. Gutman, O. P. Keifer Jr, M. E. Magnuson, D. C. Choi, W. Majeed, S. Keilholz, K. J. Ressler, "A DTI tractography analysis of infralimbic and prelimbic connectivity in the mouse using high-throughput MRI," Neuroimage, Vol. 63, No. 2, pp. 800-811, 2012.

[5] E. Calabrese, A. Badea, G. Cofer, Y. Qi, G. A. Johnson, "A diffusion MRI tractography connectome of the mouse brain and comparison with neuronal tracer data," Cerebral cortex, Vol. 25, No. 11, pp. 4628-4637, 2015.

[6] A. E. Dorr, J. P. Lerch, S. Spring, N. Kabani, R. M. Henkelman, "High resolution three-dimensional brain atlas using an average magnetic resonance image of 40 adult C57Bl/6J mice," Neuroimage, Vol. 42, No. 1, pp. 60-69, 2008.

[7] Lein ES, Hawrylycz MJ, Ao N, Ayres M, Bensinger A, Bernard A, Boe AF, Boguski MS, Brockway KS, Byrnes EJ, Chen L, Chen L, Chen TM, Chin MC, Chong J, Crook BE, Czaplinska A, Dang CN, Datta S, Dee NR, Desaki AL, Desta T, Diep E, Dolbeare TA, Donelan MJ, Dong HW, Dougherty JG, Duncan BJ, Ebbert AJ, Eichele G, Estin LK, Faber C, Facer BA, Fields R, Fischer SR, Fliss TP, Frensley C, Gates SN, Glattfelder KJ, Halverson KR, Hart MR, Hohmann JG, Howell MP, Jeung DP, Johnson RA, Karr PT, Kawal R, Kidney JM, Knapik RH, Kuan CL, Lake JH, Laramee AR, Larsen KD, Lau C, Lemon TA, Liang AJ, Liu Y, Luong LT, Michaels J, Morgan JJ, Morgan RJ, Mortrud MT, Mosqueda NF, Ng LL, Ng R, Orta GJ, Overly CC, Pak TH, Parry SE, Pathak SD, Pearson OC, Puchalski RB, Riley ZL, Rockett HR, Rowland SA, Royall JJ, Ruiz MJ, Sarno NR, Schaffnit K, Shapovalova NV, Sivisay T, Slaughterbeck CR, Smith SC, Smith KA, Smith BI, Sodt AJ, Stewart NN, Stumpf KR, Sunkin SM, Sutram M, Tam A, Teemer CD, Thaller C, Thompson CL, Varnam LR, Visel A, Whitlock RM, Wohnoutka PE, Wolkey CK, Wong VY, Wood M, Yaylaoglu MB, Young RC, Youngstrom BL, Yuan XF, Zhang B, Zwingman TA, Jones AR. Genome-wide atlas of gene expression in the adult mouse brain. Nature Vol. 445, No. 7124, pp. 168-176, 2007.

[8] Huebner NS, Mechling AE, Lee HL, Reisert M, Bienert T, Hennig J, von Elverfeldt D, Harsan LA. The connectomics of brain demyelination: functional and structural patterns in the cuprizone mouse model. Neuroimage Vol. 146, pp. 1-18, 2017.

[9] Koch S, Mueller S, Foddis M, Bienert T, von Elverfeldt D, Knab F, Farr TD, Bernard R, Dopatka M, Rex A, Dirnagl U, Harms C, Boehm-Sturm P. Atlas registration for edema-corrected MRI lesion volume in mouse stroke models. Journal of Cerebral Blood Flow & Metabolism Vol, 39, No. 2, pp.313-323, 2019.

[10] Barrière DA, Magalhães R, Novais A, Marques P, Selingue E, Geffroy F, Marques F, Cerqueira J, Sousa JC, Boumezbeur F, Bottlaender M, Jay TM, Cachia A, Sousa N, Mériaux S. The SIGMA rat brain templates and atlases for multimodal MRI data analysis and visualization. Nature communications Vol. 10, No. 1, pp. 1-13, 2019.

[11] M. Jenkinson and S.M. Smith. A global optimisation method for robust affine registration of brain images. Medical Image Analysis, Vol. 5, No. 2, pp. 143-156, 2001.

[12] M. Jenkinson, P.R. Bannister, J.M. Brady, and S.M. Smith. Improved optimisation for the robust and accurate linear registration and motion correction of brain images. NeuroImage, Vol. 17, No. 2, pp. 825-841, 2002

[13] Visser, M., Petr, J., Müller, D. M., Eijgelaar, R. S., Hendriks, E. J., Witte, M., Barkhof F, van Herk M, Mutsaerts HJMM, Vrenken H, de Munck JC, De Witt Hamer, P. C. (2020). Accurate MR image registration to anatomical reference space for diffuse glioma. Frontiers in neuroscience, 14, 585.

Reviewer 2 Report

SangJin Im et al show a deterministic tractographic pipeline for brain structural connectivity analysis using DTI data. Using the proposed analysis pipeline the authors also evaluated the white matter connectivity changes after the stroke in the rat. But there is only one animal in the stroke model group and one in the control group. How can we know the proposed analysis pipeline is robust across the animals?

  1. There is one thing that needs to be pointed out, the parameters for DTI data acquisition are totally different for the control animal and the stroke animal. (TR, TE, BW, b-value, δ, Δ, average times and total scan time). Then it is not appropriate to put the DTI results in the same figure (SUPP 1).
  2. Please provide the SNR of the DTI raw data in the manuscript.
  3. I appreciate that the authors have the schematic diagrams for data processing in the manuscript. It helps the reader a lot.
  4. Overall, the study design is straightforward, but the manuscript is not easy to follow, it needs extensive English editing.
  5. Please add the limitation/caveats of the proposed pipeline for brain structural connectivity analysis using DTI data in your discussion.

Author Response

Response to Reviewer 2 Comments

Point 1: SangJin Im et al show a deterministic tractographic pipeline for brain structural connectivity analysis using DTI data. Using the proposed analysis pipeline the authors also evaluated the white matter connectivity changes after the stroke in the rat. But there is only one animal in the stroke model group and one in the control group. How can we know the proposed analysis pipeline is robust across the animals?

Response 1: Thank you for your kind review.

A total of 7 rats were used in this study, with 1 normal rat and 6 stroke model rats. The rats stroke rat models used in the study were Sprague-Dawley (SD) models.

Our pipeline utilizes highly tested methods with extensive development, including a probabilistic tractography analysis method tested previously in studies analyzing the mouse brains of control and Parkinson’s disease models [1,2]

In this study, we initially test our deterministic tractography analysis method on a normal rat brain then apply it to a stroke model to find evidence that the structural connectivity is initially damaged in the M1 region located in the left and right hemispheres of the rat brain, but possibly restored over time.

This is a valuable study for researchers that incorporates various tools such as the SIGMA rat atlas, ANTx2 and FSL to acquire brain structure region masks, and visualize and quantify structural connectivity through DSIstudio.

To increase the reliability of this pipeline for future studies, we plan to increase the number of rats used in the experiment, combine biological and functional MR techniques, and conduct deterministic tractography studies using a variety of different disease models.

Point 2: There is one thing that needs to be pointed out, the parameters for DTI data acquisition are totally different for the control animal and the stroke animal. (TR, TE, BW, b-value, δ, Δ, average times and total scan time). Then it is not appropriate to put the DTI results in the same figure (SUPP 1).

Response 2: Based on the reviewers' opinions, the Supp 1 figure was modified to be Supp 1, which represents the DTI of the control group, and Supp2, which represents the DTI of the stroke group.

Point 3: Please provide the SNR of the DTI raw data in the manuscript. 

Response 3: In response to reviewers' comments, we added SNR information to the MRI acquisition section. Modified contents are marked in brown.

Point 4: I appreciate that the authors have the schematic diagrams for data processing in the manuscript. It helps the reader a lot. 

Response 4: We sincerely appreciate the comments of the reviewers. We hope that the pipeline presented in this study will contribute to making Rat tractography research more easily accessible.

Point 5: Overall, the study design is straightforward, but the manuscript is not easy to follow, it needs extensive English editing.

Response 5: We provided a schematic of the analysis pipeline in Figure 1 for an easier understanding of the analysis pipeline presented in the study. Although there may still be some difficulties in understanding our methods due to the utilization of 4 different softwares, we believe that it our methods has the potential to advance the accessibility of studying rat structural connectivity through tractography analysis. Minor English editing was initially done on the paper but more time will be necessary to make further adjustments.

Point 6: Please add the limitation/caveats of the proposed pipeline for brain structural connectivity analysis using DTI data in your discussion.

Response 6: We have added a limitation section before the conclusion detailing the cautions and limitations of the pipeline presented in this study based on feedback from reviewers. Added content is marked in brown.

"In this study, we presented a structural analysis and structural connectivity analysis pipeline of the rat brain by combining various existing neuroimaging analysis tools. From the results, we were able to successfully extract and segment individual ROI masks, and perform tractographic analysis. However, the study has several limitations. Due to the small number of rat samples used in the experiment, it was not possible to statistically verify the quantitative values obtained for each analysis. In the future, there is a need to increase the number of samples coupled with a longitudinal study. In addition, directly comparing pipeline results obtained using normal and stroke models may be difficult because the image data acquisition parameters of the normal rat and stroke rat models are different. In future studies, all images of both normal and stroke models will be obtained using the same parameters. Another limitation lies in the registration method used to obtain the desired structural region mask in the pipeline presented in the study. It is difficult to confidently measure volumetric change of mouse brain structures, due to possible errors that may have occurred due to the quality of the acquired image or the registration algorithm. In addition, there is a limitation to the careful evaluation of the registration method used to measure accuracy. In this study, although the SIGMA atlas has been used previously for accurate structural segmentation of the rat brain, there are limitations in how accurate the registration tools and qualitative visual inspection can be. Finally, the results of tractographic analysis can be influenced by various factors such as the quality of the image data and the parameters used to trace the fiber. It is also difficult to interpret certain characteristics of fibers because certain aspects of the generated fibers such as the projection direction cannot be distinguished."

1] Shim, J. H., Sang-Jin Im, A., Kim, Y., Kim, Y. T., Kim, E. B., & Baek, H. M. (2019). Generation of mouse basal ganglia diffusion tractography using 9.4 T MRI. Experimental neurobiology, 28(2), 300.

[2] Kim, A. Y., Oh, C., Im, H. J., & Baek, H. M. (2020). Enhanced Bidirectional Connectivity of the Subthalamo-pallidal Pathway in 6-OHDA-mouse Model of Parkinson’s Disease Revealed by Probabilistic Tractography of Diffusion-weighted MRI at 9.4 T. Experimental neurobiology, 29(1), 80.

Round 2

Reviewer 1 Report

Thank you for the explanation, but I still have doubts.

First, the atlas method is not the only approach to segmenting brain structures. You should refer to other methods and justify the use of just such.

Second, in the answer to question 1 it was stated that accurate results were obtained for the segmentation of the rat brain. In turn, the answers to question 2 show that the accuracy of segmentation was assessed only visually (no quantitative assessment was performed), so it is probably difficult to say that these results are accurate. This is therefore in contradiction with the declaration in the Introduction about the high accuracy of the proposed method.

Please address these two points. 

Author Response

Response to Reviewer 1 Comments

Point 1: Thank you for the explanation, but I still have doubts.

First, the atlas method is not the only approach to segmenting brain structures. You should refer to other methods and justify the use of just such.

Response 1: Thank you for your kind review.

While it is true that there are other methods for segmenting, such as manual segmentation and machine learning, most segmentation methods rely on tracing and adjusting ROIs based on outlines visualized by differences in MRI image contrasts. However, manual segmentation of the rat brain is an arduous, time-consuming task, riddled with inconsistencies due to the lack of defining lesion boundaries of brain structures, particularly in low resolution images [1,2]. Machine learning based segmentation tools also rely on lesion boundaries to train and may be inconsistent if used to define structures of MRI images with protocols different than the ones the algorithm was trained on. As such, use of semi-automatic atlas-based segmentation is a commonly used compromise even in human brain studies. We also highlight the lack of toolboxes for automatically segmenting rat brain structures and believe that our methods can help reduce burdens of human workload and improve reproducibility of segmentation.

Point 2: Second, in the answer to question 1 it was stated that accurate results were obtained for the segmentation of the rat brain. In turn, the answers to question 2 show that the accuracy of segmentation was assessed only visually (no quantitative assessment was performed), so it is probably difficult to say that these results are accurate. This is therefore in contradiction with the declaration in the Introduction about the high accuracy of the proposed method.

Response 2: The expression that accurate results were obtained from the manuscript has been deleted by collecting the opinions of reviewers. For example, the phrase "which has recently shown high accuracy" was deleted from the introduction section, and the qualitative nature of our evaluation was added to the limitations.

[1] Iglesias, J. E., & Sabuncu, M. R. (2015). Multi-atlas segmentation of biomedical images: a survey. Medical image analysis, 24(1), 205-219.

[2] Gutman, D. A., Keifer Jr, O. P., Magnuson, M. E., Choi, D. C., Majeed, W., Keilholz, S., & Ressler, K. J. (2012). A DTI tractography analysis of infralimbic and prelimbic connectivity in the mouse using high-throughput MRI. Neuroimage, 63(2), 800-811.

Reviewer 2 Report

It may mislead the readers eventhough the authors split the supplimentary figure to 2. I will strongly suggest to acquire the DTI data using exactly the same parameters in namal rats. The group size should be at least 3.

Please be aware that the SNRs for the DTI data are low (25, 16, 18...) in this study, the recommended SNRs should be more than 40. I will suggest to add this point to the discussion or the limitation section.

Author Response

Response to Reviewer 2 Comments

Point 1: It may mislead the readers eventhough the authors split the supplimentary figure to 2. I will strongly suggest to acquire the DTI data using exactly the same parameters in namal rats. The group size should be at least 3.

Response 1: Thank you for your kind review.

Tabs in Supplementary Figure 1 and Figure 2 have been completely separated to reduce confusion for the reader. In addition, we added "DTI data FA maps of normal rat and stroke rat models with different parameters are presented in Supplementary Figures 1 and 2." in the methods section.

In addition, this study was a preliminary study, utilizing experimental data with varying protocols to test our pipeline. In future studies, we plan to increase the number of Rats used in the experiment and acquire DTI images with the same parameters to increase statistical reliability.

Point 2: Please be aware that the SNRs for the DTI data are low (25, 16, 18...) in this study, the recommended SNRs should be more than 40. I will suggest to add this point to the discussion or the limitation section.

Response 2: In response to reviewers' opinions, limitations regarding SNR were added to the Discussion section.

Modifications and additions were highlighted in yellow.
